youth; mental health; perinatal; Africa; LMIC

**Corresponding author:**
Wezi Mhango;
Emails: wm90@sussex.ac.uk;
wmhango@unima.ac.mw

$Joint senior authors.

# "I felt I needed help, but I did not get any": A multiple stakeholder qualitative study of risk and protective factors, and barriers to addressing common mental health problems among perinatal adolescents in Malawi

Wezi Mhango[1,2] , Daniel Michelson[1,3$] and Darya Gaysina[1$]

[1]School of Psychology, University of Sussex, Brighton, UK; [2]Department of Psychology and Medical Humanities, University of Malawi, Zomba, Malawi and [3]Department of Child and Adolescent Psychiatry, Institute of Psychiatry, Psychology and Neuroscience, King's College London, London, UK

## Abstract

Common mental health problems (particularly depression and anxiety) are common among adolescents during the perinatal period. Previous research has identified the distinctive needs of this group and called for contextually appropriate psychosocial interventions. The current study conducted in Malawi aimed to explore risk and protective factors for common mental health problems, and barriers to accessing mental health care, among perinatal adolescents, to develop a contextually relevant intervention for preventing and treating perinatal depression and anxiety. An exploratory qualitative study was conducted in antenatal and postnatal clinics in Zomba district, Malawi in January–March 2022. In-depth individual interviews were completed with perinatal adolescents aged ≤19 ($n = 14$); their family members ($n = 4$); and healthcare workers ($n = 8$). Interview data were subjected to thematic framework analysis. Data were organised around two themes: "psychosocial risk and protective factors" (potential causes of common mental health problems among adolescents); and "health care services" (maternal and mental health services available, and adolescents' experiences of using these services). Interventions need to go beyond targeting symptoms of depression and anxiety to addressing the wider contextual risk factors and barriers to care at the different socioecological levels.

## Impact statement

Common mental health problems, particularly depression and anxiety among adolescents in the perinatal (prenatal and postpartum) period, are major public health concerns with several detrimental outcomes for both the adolescent and the foetus or infant. The prevalence rates are particularly higher in low- and middle-income countries (LMICS), including Sub-Saharan Africa. However, interventions addressing these conditions in this region are scarce. This study investigated the risk and protective factors, and barriers to addressing perinatal depression and anxiety among adolescents in Malawi. It is the first phase of a larger programme of study aimed to develop a contextually appropriate psychosocial intervention for perinatal depression and anxiety among adolescents in Malawi. The findings of the study, drawing on qualitative interviews with 26 participants from three stakeholder groups (perinatal adolescents, their family members, and healthcare workers), have important practice implications. First, interventions need to go beyond targeting symptoms of depression and anxiety to address the wider contextual risk factors while enhancing protective factors at the different socioecological levels. Also, considering the scarcity of mental health specialists, current healthcare systems need to be strengthened by training non-specialist health workers to detect and manage perinatal mental health problems among adolescents.

## Introduction

Common mental health problems, particularly depression and anxiety, are common in the perinatal period: during pregnancy and 12 months after giving birth, with an estimated global prevalence of 11.9% and 20.7%, respectively (Woody et al., 2017). Perinatal risks of depression and anxiety are especially high in low- and middle-income countries (LMICs) where pregnant and postpartum women are exposed to an array of psychosocial risk factors including low education attainment, poverty, and food insecurity (Gelaye et al., 2016; Tomlinson et al., 2018). Many of these risks are compounded for perinatal adolescents, who are more likely to

experience a lack of social support, stigma, and other forms of social and economic disadvantage (Rubertsson et al., 2014; Munthali and Kok, 2016; Recto and Champion, 2017; Field, 2018; Hymas and Girard, 2019). However, the lack of specialist perinatal services and under-resourced mental health care more generally, means that most perinatal mental health problems go undetected or untreated in LMICs (Udedi, 2016).

Malawi, a low-income country in southeastern Africa (population c. 20 million), has one of the world's highest adolescent birth rates. The birth rate among women aged 15–19 is estimated at 136 births per 1,000 women (National Statistics Office (Malawi) and ICF, 2017) compared with a global birth rate of 42.5 births per 1,000 women in the same age cohort (WHO, 2022). There is growing recognition of the importance of healthcare services specifically designed to address mental health needs of perinatal adolescents (WHO, 2012; Daley et al., 2013; Field et al., 2020). Yet, implementation is restricted by lack of resources (Udedi, 2016), competing health priorities (Government of Malawi, 2017) and limited local evidence.

A recent systematic review on adolescent perinatal mental health in Sub-Saharan Africa and South Asia did not find any published evidence of targeted perinatal mental health interventions for adolescents in Africa (Palfreyman and Gazeley, 2022). Similarly, in a systematic review of preventive psychosocial interventions for mental health in perinatal adolescents (Laurenzi et al., 2020), none of the 17 included randomised controlled trials of perinatal mental health interventions were from LMICs. There is also a lack of formative qualitative evidence to guide development of adolescent-focused perinatal interventions. The available research in Malawi (Ng'oma et al., 2019) focused on older women and found that perinatal depression affected the daily functioning of women, suggesting the need for psychosocial interventions to address identified difficulties. The study also found that women used idioms to describe distress, hence, researchers and service providers need to use contextually recognised terminology or idioms for the detection and management of mental health problems to increase understanding and acceptability.

Local terms and idioms are used for emotional distress and mental disorders depending on the linguistic, cultural, and regional context (Stewart et al., 2015). Researchers (Kaiser et al., 2015; Cork et al., 2019) argue that local terms influence how people understand and respond to mental health conditions. As such, it is important to identify the terminology that is used in different communities and use this to facilitate culturally relevant interventions and reduce health disparities (Ng'oma et al., 2019; Jumbe et al., 2022).

This exploratory qualitative study (Hunter et al., 2019) is the first phase of a larger programme of research that aims to develop and evaluate a contextually appropriate, transdiagnostic intervention for adolescents experiencing perinatal depression and anxiety in Malawi. There is strong global interest in transdiagnostic approaches to early intervention and treatment, reflecting evidence for frequent comorbidity, overlapping genetic and environmental risk factors, and scalability of parsimonious practice elements (Falah-Hassani et al., 2017; Singla, 2021). In the present study, we aim to address two questions: (1) What are the contextual risk and protective factors for common mental health problems (depression and anxiety) among perinatal adolescents in Malawi?, and (2) What are the demand and supply side barriers to accessing help for common mental health problems (depression and anxiety) among perinatal adolescents through current service provision?

## Methods

### Study design and setting

We carried out a cross-sectional qualitative study using data from stakeholder interviews, completed from January to March 2022. This design enabled the researchers to collect detailed descriptive data from different participants within the same specific point in time (Wang and Cheng, 2020). The study has been reported in line with the Consolidated Criteria for Reporting Qualitative Research (COREQ) (Tong et al., 2007).

The study was conducted at Matawale and Naisi primary health centres, representing urban and rural localities in the Zomba district (population 851,737; National Statistical Office (Malawi), 2019), southern Malawi. According to the most recent Malawi Demographic and Health Survey, 34.9% of women aged 15–19 in the Zomba district have experienced childbirth. This is higher than the national average rate of 29% (National Statistical Office (Malawi) and ICF, 2017). Although there are currently no prevalence statistics for depression and anxiety among perinatal adolescents in Zomba district, research has shown that becoming a parent at a young age can put adolescents at risk of developing common mental disorders such as depression and anxiety (Agnafors et al., 2019).

### Participants

The sampling frame was all adolescents attending antenatal and under-five clinics at one rural (Naisi) and one urban (Matawale) health centre in Zomba district Malawi over a period of 8 weeks, family members of consenting adolescents, and maternal and mental health providers working at these clinics.

(1) Eligible perinatal adolescents were those aged ≤19 years, who were attending antenatal or postnatal clinics (N = 14). Consecutive sampling was used. This sampling technique allows for the recruitment of all eligible participants until the required sample size is achieved, thereby reducing sampling bias (Thewes et al., 2018). A nurse in charge provided basic verbal information about the study to adolescents in waiting rooms. Interested adolescents were directed to the researcher who provided further details of the study verbally and in a written information sheet.

(2) Eligible family members (partners, parents or guardians, siblings, in-laws) were those aged 18+ years, living in the same household as an index perinatal adolescent who had already consented to be in the study (N = 4). This group was sampled from among family members who accompanied a consenting adolescent to the antenatal or postnatal clinics.

(3) Eligible healthcare workers were those who had worked at the respective health centre for at least 12 months and provided maternal and/or mental health care to perinatal adolescents (N = 8). This included workers employed directly by the health centre, and employees of non-governmental organisations (NGOs) who provide on-site psychosocial support to youth at the health centres. A snowball method was employed where participants were asked to provide contact details for suitably experienced colleagues. Potential participants were either approached in person within the health centre, or through telephone.

In total, we interviewed 26 participants. All the participants we approached consented to participate in the study. None of the participants withdrew from the interview process. The final sample size was determined by continuously analysing the data to determine whether adequate information power had been achieved – information power indicates that the actual number of participants

needed for a study is determined by the amount of information the sample holds that is relevant for the study (Malterud et al., 2016).

### Measures

Before the interviews, we assessed symptoms of depression and anxiety using Chichewa versions of the Edinburgh Postnatal Depression Scale (EPDS), Generalised Anxiety Disorder 7-item scale (GAD-7) and the Self Reporting Questionnaire (SRQ) (Stewart et al., 2013). The SRQ and EPDS have been validated in Malawi with high internal validities of α = 0.825 and α = 0.904, respectively (Stewart et al., 2009; Stewart et al., 2013). The GAD-7 has not been validated in Malawi. However, it has been validated among perinatal populations in other LMICs (Mughal et al., 2020). The resulting scores were used to characterise the sample and not as the basis for eligibility screening. Participants with moderate to high levels of symptoms of depression and anxiety were signposted to relevant institutions for further assessment and management.

Semi-structured interview topic guides used widely used and understood terminology. During recruitment, the aims of the study were presented to participants as relating to understanding more about "maganizo angwiro" (which translates as "mental health") during the perinatal period. Within this broad remit, we were interested in exploring widely used terminology or idiomatic descriptions related to both symptoms and syndromes that would otherwise be termed "anxiety" and "depression".

In Malawi, the Chichewa medical or technical terminology for depression is "matenda okhumudwa" while "matenda a nkhawa" is used to refer to anxiety disorders. However, researchers (Kutcher et al., 2019; Jumbe, 2021) have suggested the use of terminology that is contextually used and that participants in the study can relate to. Hence, to ensure that the terminology used in this study is relevant to the study participants and context, we used case vignettes to orient participants to the array of symptoms associated with anxiety and depression, and to clarify the shared vernacular or local terminology for these.

Topic guides for perinatal adolescents and family members began with two case vignettes adapted from a study conducted in rural communities in Vietnam (Abrams et al., 2016) which illustrated respective cases of postnatal depression and prenatal anxiety, reflecting criteria from the Diagnostic and Statistical Manual of Mental Disorders (4th ed., text rev.; DSM-IV-TR; American Psychiatric Association, 2000). These vignettes reflected characters that were adolescents in the perinatal period. Adolescents and family members were asked to interpret the vignettes by answering questions from the Short Explanatory Model Interview (SEMI) (Lloyd et al., 1998). This included asking participants whether they felt the person is experiencing an illness, what term(s) best describes the illness (or problem), what could be the causes, and what treatment options or supports may be needed.

Although some participants summarised the case presentations as "kuganiza kwambiri" (meaning "thinking too much"), most respondents used the term "matenda a nkhawa" (meaning "anxiety disorder" or in more literal terms "illness of anxiety"). Although the term "matenda a nkhawa" can be seen as relating more to anxiety disorder, it was noted that participants used this term to describe a spectrum of anxiety and depression symptoms and did not differentiate any further. The common use of this term in a transdiagnostic manner is consistent with previous studies (Kutcher et al., 2019; Jumbe, 2021) and consultations carried out with healthcare workers prior to data collection – health workers indicated that although they are aware of the different medical terminology for depression and anxiety, they usually use the term "matenda a nkhawa" to refer to both depression and anxiety as that is the term that is most commonly used and understood in the communities. Therefore, for the purposes of this study, the term "matenda a nkhawa" was used consistently throughout. To ensure that the intent of young people's narratives was captured accurately, we have used the vernacular terms whenever possible. These include "nkhawa" meaning ("worry") and "kukhumudwa" (meaning "feeling low" or "feeling sad").

Additional topic guide questions related to stakeholders' perspectives of adolescents' experience of pregnancy and the postpartum period; understanding and awareness of symptoms of perinatal depression and anxiety: and formal and informal support systems available for adolescents in the perinatal period.

### Procedures

Written informed consent was obtained from all adolescents aged 18+ years. Assent was obtained from participants aged <18 years backed by consent from a parent/guardian. Married participants aged <18 years are considered emancipated minors in Malawi and provided consent independently. Information sheets detailing study aims, benefits, and potential risks were provided to all participants (and a parent/guardian where appropriate) in both English and Chichewa.

Interviews lasting between 30 and 60 min were conducted face-to-face in a private space within the health facility to ensure confidentiality. All interviews were audio recorded. The interviews were conducted either in English or Chichewa (depending on the participant's preference) by WM who is bilingual and has prior training and experience in conducting qualitative interviews. Recorded interviews were transcribed verbatim. Chichewa transcripts were subsequently translated to English by WM prior to data analysis.

### Data analysis

The Framework Analysis method (Gale et al., 2013) was used to analyse the data. This involved transcription, familiarisation, coding, developing an analytical framework, indexing, charting, and interpreting the data. We utilised both deductive coding (following logically from our research questions) and inductive data-driven coding. WM and DG independently coded three transcripts line by line. The codes were compared and refined, and a code list was agreed upon and organised into thematic categories by all researchers. This working framework was then applied to the rest of the transcripts by WM with feedback from DG and DM. Related codes were merged, and final themes and subthemes were generated. Illustrative quotes were selected within each subtheme, and the supporting narrative included consideration of areas of divergence and convergence within and across stakeholder groups.

Formally assessing validity and reliability was not appropriate to our mode of qualitative analysis where we were exploring themes rather than focusing on specific categorical counts. Nevertheless, we ensured validity in the following ways: (i) we triangulated findings from three stakeholder groups to ensure credibility and authenticity; (ii) the findings were discussed in a group and analysed by multiple researchers to avoid bias and ensure integrity; and (iii) the researchers did not have a predetermined theoretical perspective. This enabled the researchers to critically appraise the different aspects of the research.

All interviews were conducted by WM who is a psychologist and a mother to a young child. WM also lives in the context where the study took place. Her prior understanding of the context and health systems allowed her to easily build rapport and gain the trust of the research participants. In addition, holding a "shared experience" position with the adolescents in the perinatal period enabled WM to ask relevant follow-up questions and reduce distance during the interviews. To maintain objectivity, the research team also comprising a clinical psychologist (DM) and a developmental psychologist (DG) met regularly at each step of the research process to formulate research questions and topic guides and discuss and analyse the findings of the study. In addition, members of the research team were from different research backgrounds, hence we did not have any pre-existing theories that might have affected the interpretation of the results.

## Results

Characteristics of study participants are summarised in Table 1. Half of the adolescent participants had medium to high depressive scores, and 64% had low to mild symptoms of anxiety.

### Theme 1: Psychosocial risk and protective factors

This theme encompassed the psychosocial factors that were identified as potential causes and protective factors of common mental health problems for adolescents in the perinatal period.

#### (Lack of) supportive relationships

The negative psychological effects of diminished social support (and conversely, the protective effects of positive support) were consistently identified by all stakeholder groups. Social support was described in terms of both emotional and instrumental support from parents, partners, peers, and other community members.

> The relationship (with my partner) is there but it is not as it was because his parents told him not to assist me in any way since my parents refused that I should go and stay with him… I felt hurt, but I just let it go. (Pregnant adolescent 1, 17 years old, Naisi)

> My parents support me. My partner also supports me. They give me soap, lotion, they cook for me. My baby is now two weeks old, but my mother still does most of the things for me. So, I don't have "nkhawa". (Postpartum adolescent 1, 18 years old, Matawale)

The important role of social support was echoed by family members and service providers

> Some think a lot when they don't have a partner… This mostly happens when the father of the child denies responsibility. So, the girl thinks of where she will get support for the child once he or she is born. (Sister, Naisi)

#### Body and birth worries

Adolescents described worries related to body changes (e.g., weight gain) that were a result of pregnancy, and ultimately the process of labour.

> If I make a sudden trip, she has "nkhawa". She thinks I have gone to see another woman and I am running away from her because she is pregnant and she is gaining weight… You will just notice that she starts keeping to herself and she is very quiet, like "kukhumudwa". (Partner, Naisi)

**Table 1.** Demographic characteristics of participants

| Characteristics | N (%) |
|---|---|
| *Perinatal adolescents* | |
| **Age of participants, years** | |
| Mean (SD) | 17.3 (0.8) |
| Range | 16–19 |
| **Perinatal period** | |
| Prenatal | 9 (64.3) |
| Postpartum | 5 (35.7) |
| Total | 14 (100) |
| **Marital status** | |
| Single | 3 (21.4) |
| Married before pregnancy | 4 (28.6) |
| Married during/after pregnancy | 7 (50.0) |
| **Education level** | |
| Primary | 10 (71.4) |
| Secondary | 4 (28.6) |
| **Current education status** | |
| In school | 0 (0.0) |
| Temporarily dropped out | 8 (57.1) |
| Permanently dropped out | 6 (42.9) |
| **Source of income** | |
| Support from family | 6 (42.9) |
| Piece work | 8 (57.1) |
| *Family members* | |
| **Categories** | |
| Partners | 2 (50) |
| Sister | 1 (25) |
| Sister-in-law | 1 (25) |
| Total | 4 (100) |
| **Sex** | |
| Male | 2 (50) |
| Female | 2 (50) |
| **Age, years** | |
| Mean (SD) | 26 (8.8) |
| Range | 18–38 |
| *Service providers* | |
| **Categories** | |
| Nurses/midwives | 3 (37.5) |
| Health surveillance assistants | 2 (25) |
| Clinical officer | 2 (25) |
| Social worker | 1 (12.5) |
| Total | 8 (100) |
| **Sex** | |
| Male | 3 (37.5) |

*(Continued)*

**Table 1.** (*Continued*)

| Characteristics | *N* (%) |
|---|---|
| Female | 5 (62.5) |
| Total | 8 (100) |
| **Age, years** | |
| Mean (SD) | 33.8 (5.6) |
| Range | 28–44 |
| **Years of service** | |
| Mean (SD) | 7.3 (5.9) |
| Range | 2–17 |
| ***EPDS, SRQ and GAD-7 scores for adolescents*** | |
| **EPDS** | |
| Mean score (SD) | 6.7 (6.5) |
| Low (0–6) | 7 (50.0) |
| Medium (7–11) | 3 (21.4) |
| High (>11) | 4 (28.6) |
| **SRQ** | |
| Mean score (SD) | 6.4 (4.9) |
| Low (0–4) | 7 (50.0) |
| Medium (5–7) | 1 (7.1) |
| High (≥8) | 6 (42.9) |
| **GAD-7** | |
| Mean score (SD) | 6.7 (5.8) |
| Minimal (0–4) | 5 (35.7) |
| Mild (5–9) | 4 (28.6) |
| Moderate (10–14) | 4 (28.6) |
| Severe (≥15) | 1 (7.1) |

Sometimes even when people told me not to worry, I wouldn't listen because that fear of giving birth had already been planted in me. My peers would say, you are going to see that this is not child's play, you will see. (Postpartum adolescent 1, 18 years old, Matawale)

#### Role transitions

Adjustment to the idea of becoming a mother was particularly difficult among adolescents with unplanned pregnancy.

When some girls give birth, they feel like at that time they have lost value or are worthless. They cannot do anything useful anymore. They think when they have given birth, everything has come to a standstill because they have to focus on taking care of the baby. (Pregnant adolescent 4, 18 years old, Naisi)

#### Stigma towards teen pregnancy

This subtheme reflected how society and healthcare workers perceive teenage pregnancies. Pregnant adolescents were considered a "loose and bad influence" on their peers. This was more common for younger adolescents or if the child was conceived out of wedlock and the partner denied responsibility. Those who were married either before getting pregnant or following pregnancy were viewed more favourably by their community regardless of their age.

Most of them were laughing and saying a lot of bad things. Some even said, "as young as you are you have already started being a prostitute?"… There were times I felt sad, but I decided to ignore what they are saying and avoid them… Now that I have a baby, they don't say anything. Maybe because I am now married. (Postpartum adolescent 3, 16 years old, Matawale)

Negative attitudes were also reflected in family members' and health workers' reactions to pregnant adolescents.

What I would say to her is "listen, we told you not to move around with boys. Now, this is the result. You have destroyed your future… You shouldn't worry about anything because these are the consequences of your actions." (Sister-in-law, Matawale)

We first ask her, "what were you thinking? Why did you get pregnant?" Because nowadays there are a lot of contraceptives. Anybody can get them, whether young or old, we give them to everybody. So, we ask her, "were there no condoms? Why didn't you come to get contraceptives?" Then at the end, we encourage her and tell her that she will deliver properly, and she can return to school once she delivers. (Nurse/Midwife, Naisi)

#### School dropout

School dropout was common among adolescents in the perinatal period. Reasons for dropping out included being ashamed to attend school while pregnant, being unable to balance being pregnant and attending school, and receiving negative remarks from peers and other community members.

When I was pregnant, I just decided to drop out because I felt ashamed. (Postpartum adolescent 5, 17 years old, Matawale)

Although some adolescents were willing to return to school after delivery, they expressed concern over delaying while their peers progressed to other classes.

When I give birth, I will return to school. The school allows you to go back, there are no penalties. My only "nkhawa" is that my friends will have moved forward, and I will be behind…I feel sad when I see my friends going to school. (Pregnant adolescent 7, 17 years old, Naisi)

On the other hand, some adolescents who were married reported that they could not return to school as they now had the responsibility of taking care of their babies and their new homes.

Now I am taking care of the baby since he is still young, but I am not sure if I will go back to school. I am married now; I have to take care of my home. (Postpartum adolescent 3, 16 years old, Matawale)

#### Financial struggles

It was reported that adolescents who lacked support from their parents and/or partners often worried about being able to source materials such as wrappers and basins in preparation for birth. In addition, they worried about not being able to find resources to take care of their babies once they are born. Adolescents also reported being sad about not being able to afford the things they wanted such as buying certain food that they were craving.

I try to be happy but sometimes when I think about where we will get money to take care of the baby, I get worried. My partner doesn't have a stable job. So, it is hard not to be sad. (Pregnant adolescent 9, 18 years old, Naisi)

When the baby is born the baby needs this and this. But now, many of these adolescents might be in school. Some even in primary. This means the responsibility mainly lies on the parents who might not even have much. Worse still when the husband or the boyfriend has denied responsibility. All those things can result into depression. (Clinical officer, Matawale)

### Theme 2: Health care services

This theme related to participants' experiences of maternal and mental health services, as well as barriers to accessing these services.

### Experiences of maternal health care

Maternal care mainly included providing nutritional and family planning information and conducting routine physiological antenatal and postpartum checks for mother and baby. However, information on perinatal mental health was not provided.

> We don't provide information about mental health at any point. We only talk about it if we notice that a person looks unhappy, that is if at all you notice. We will then talk with that person. (Community Midwife Assistant, Naisi)

Although most adolescents reported being satisfied with the services provided, there were mixed views regarding attending antenatal clinics where older women were also service users. While some adolescents felt that they could learn from older women since they may be more experienced, others reported that they felt uncomfortable as the older women would look at them in a judgemental way. This was corroborated by some health workers.

> When I was pregnant, we did not receive any special care. We attended the antenatal clinic with everybody else, older, or younger…Some older women would look at me in a judgemental way. That made me uncomfortable. (Postpartum adolescent 2, 16 years old, Matawale).

> We are assisted together regardless of age…I don't mind because we learn a lot of things from the older women. (Pregnant adolescent 4, 18 years old).

> I think if we had these youth-friendly health services specifically for antenatal it would even reduce the stigma for the adolescents because when they are combined with older women they are judged a lot. (Clinical Officer, Matawale)

### Barriers to maternal health care

Barriers included personal and health worker attitudes, myths surrounding pregnancy, and a lack of drugs such as iron tablets. Although most adolescents reported that they lived far from the health facilities, they reported that they were still able to attend the routine clinics.

> Some girls are just shy to come… Some worry that if a lot of people see them pregnant, the pregnancy will go away. So, they just stay at home, they do not come for antenatal clinics. (Health Surveillance Assistant, Naisi)

### Experiences of mental health care

Although perinatal adolescents and their family members felt the hospital would be the best place to seek help for common mental health problems, none of the adolescents had sought any mental health care. However, some adolescents reported that some nurses would give them advice related to mental health if they noticed that an adolescent looked distressed.

> There was a day I quarrelled with my partner at home, and I was crying during the antenatal clinic. So, one nurse called me and asked me what was wrong, and I couldn't say anything. Then she said I shouldn't be thinking a lot because it can affect the baby. When I am sad, the baby can feel it and will also be sad. We did not speak a lot, but I felt seen. (Pregnant adolescent 9, 18 years old, Naisi)

This was corroborated by healthcare workers who stated that perinatal adolescents rarely sought mental care, even if they seemed distressed. In terms of diagnosis and management of cases, it was reported that no screening tools were used routinely. Health workers identified cases by simply looking at how the person was looking or behaving and provided some counselling to encourage the adolescent to remain positive, usually without any follow-up. In some instances, the cases were referred to clinical officers (non-physician clinicians with basic training in diagnosing and treating common medical conditions) who provide basic psychosocial support to people living with HIV/AIDS at the targeted health centres. The officers would then assess the clients and provide psychosocial counselling. More severe cases were referred to the psychiatric hospital.

> We identify these patients as we are interacting with them or when our friends from the maternity section refer them to us…we do a history, we do an examination by asking questions, we can tell that this one is dealing with depression or just anxiety. (Clinical officer, Matawale)

### Demand barriers to mental health care

Several demand-side barriers were identified, and these were categorised into knowledge-related factors and personal- and community-related factors. Knowledge-related factors included adolescents not recognising that they have a mental health problem and the lack of information on where to get help. Some participants attributed symptoms of common mental health problems to witchcraft and suggested getting spiritual help such as going for prayers or visiting a traditional healer.

> Most people don't know that what they are experiencing is a mental health problem, they think it is just one of those things to do with pregnancy. So, they will just stay home and not go to the hospital for help. (Partner, Naisi)

> It could be witchcraft…when you go to the hospital, they cannot find any illness…some go for prayers, others visit "asing'anga" (traditional healers). (Pregnant adolescent 7, 17 years old, Naisi)

Personal- and community-related factors captured personal and community attitudes towards help-seeking for mental health problems. Although most perinatal adolescents and their family members reported that seeking medical attention would be most helpful to address common mental health problems, they reported that adolescents mostly utilised informal coping mechanisms. These included isolating themselves, sleeping, and ignoring negative feelings with the hope that the feelings would eventually go away. Perinatal adolescents also reported that they utilised help from others by talking to someone (friends, relatives, older women in the community), playing with children, and being in the company of others to distract themselves from feeling sad or worried.

> I felt I needed help, but I did not get any…I wanted to ask why my heart was beating fast since my BP was okay, but I just kept procrastinating. Deep down I knew it only starts when I have "nkhawa" about something, but I did not want to be seen as being weak. After all, everyone has their own "nkhawa". It is normal. (Pregnant adolescent 1, 17 years old, Naisi).

> I have spoken to some of the women in my community about my problems and they advise me not to think a lot because you can become suicidal when you think too much. But I rarely talk to them. I don't want them to see me as someone who is always complaining. Now when I am sad, I just sleep to forget my problems. (Pregnant adolescent 9, 18 years old, Naisi).

### Supply barriers to mental health care

This subtheme captured health system-related factors. Health workers reported that they were unable to provide mental health

services due to the lack of trained personnel and private spaces where the services could be provided. They recommended that, since there is a shortage of mental health professionals, all health workers who provide maternal care should be trained in the identification and management of cases of common mental health problems.

> I think in terms of capacity, as a health facility I can say we are paralysed in terms of mental health or psychosocial counselling staff because in these issues you need proper training and you need to practice regularly… I think if all the healthcare workers, in this case, those providing maternal care, can receive training on how to effectively identify and manage cases it would be great. (Clinical officer, Naisi)

## Discussion

In this study, lack of social support, body and birth worries, role transitions, stigma from the community and health workers towards teen pregnancy, school dropout, financial struggles were identified as risk factors for common mental health problems among perinatal adolescents. While most perinatal adolescents in this study were satisfied with the maternal care received, there were mixed views regarding the lack of adolescent-friendly antenatal and postnatal clinics. Demand and supply barriers to accessing maternal mental care were also reported. These included lack of awareness of symptoms, lack of trained personnel for mental health service provision, and personal and community attitudes towards help-seeking.

It is notable that although we were not specifically selecting participants based on their reported levels of depressive or anxiety symptoms, about half of the participants had moderate to high scores on the EPDS and SRQ. This suggests that there is a need to develop an intervention to reduce symptoms of depression and anxiety among this population.

Similar to findings from previous studies (van Heyningen et al., 2016; Recto and Champion, 2017; Chorwe-Sungani and Chipps, 2018; Ajayi et al., 2023) lack of social support (from partner or parents) was reported as a major source of distress among perinatal adolescents as it left them in a position where they were unable to provide for themselves and their babies, whereas presence of social support (instrumental and emotional) was the only reported protective factor. Social support can be helpful in helping perinatal adolescents transition to the role of a mother by building resilience (Easterbrooks et al., 2016). In addition, emotional support from a significant other can help address low self-esteem that may result from various factors, including body changes, rejection by the partner, and stigma from peers, healthcare workers, and society (Ellis-Sloan, 2014; Recto and Champion, 2017).

Previous quantitative studies (Brar et al., 2020) have highlighted intimate partner violence (IPV) as a risk factor for perinatal common mental problems among adolescents. However, in this sample, none of the stakeholder groups reported any form of IPV that adolescents may be experiencing. This may be because some of the adolescents were single and no longer in contact with their partners, therefore less likely to experience IPV. In addition, most of the adolescents who were married reported that they were not living with their partners. Rather, they were either living with their parents or their partner's parents as they were in a better position to look after the adolescent during pregnancy and immediately after childbirth. Evidence has also shown that cases of IPV are often unreported due to social stigma or women being afraid of making the situation worse for themselves (WHO, 2021). This might have been the case in this study. It is possible that adolescents might have experienced IPV although did not disclose their experiences during the interview process. This warrants further qualitative research to understand perinatal adolescents' experiences of IPV.

Other studies have reported that adolescent mothers experience stigma in the community as well as the clinic (from service providers and older women attending antenatal clinics) due to the cultural values and negative stereotypes associated with teen pregnancy, especially if it is out of wedlock (Ellis-Sloan, 2014; Munthali and Kok, 2016; Kola et al., 2020). In the current study, adolescents reported facing stigma mostly from their peers and family members, especially if they were unmarried. As a result, most adolescents stated that they either opted to move in with the father of their child or were forced by their family to get married to avoid bringing shame to the family. In addition, most adolescents in this study reported that they dropped out of school due to the shame they felt and also because they felt they had to take on the new role of being a wife and mother. This was often accompanied by feelings of sadness and regret, which is consistent with previous research (Palfreyman and Gazeley, 2022).

Unplanned pregnancy was reported as a potential risk for common mental health problems. This is in line with studies that found that adolescents who had an unplanned pregnancy were at higher risk of developing depression and anxiety (Radoš et al., 2018; Nicolet et al., 2021). Furthermore, there was a consensus among all the participant groups reported that most adolescents are likely to experience "nkhawa" mainly due to the fear of childbirth, which is a common risk factor for perinatal anxiety, especially among first-time mothers (Rubertsson et al., 2014; Field, 2018).

While most adolescents reported being satisfied with the maternal health care that they received, mental health services were rarely experienced by adolescent participants in our study. This can be considered as both a demand and a supply barrier. Consistent with other studies (Nakku et al., 2016; Kotoh et al., 2022), adolescents favoured informal coping strategies such as talking to a friend over formal help-seeking, although they recognised that the hospital may be the best place to seek mental health care. This demand barrier was mainly due to perinatal adolescents' unwillingness to seek help so as not to seem weak or different from their peers. In addition, due to widespread beliefs about witchcraft among community members, mental illness was attributed to being bewitched. Therefore, one was likely to be taken to the witch doctor or traditional healer for treatment as opposed to formal health care (Nakku et al., 2016). The lack of formal help-seeking among participants in this study could also reflect lack of need as participants were not recruited based on elevated symptoms. On the supply side, health workers reported that besides perinatal adolescents not seeking formal care, the health workers did not have the capacity (in terms of training and infrastructure) to adequately address mental health problems. In line with previous research (Chowdhary et al., 2014) health care workers suggested that training non-mental health specialists such as nurses, midwives, and Health Surveillance Assistants in the identification and management of mental health problems can help improve mental health service provision.

Although previous studies (WHO, 2012; Field et al., 2020) suggest the need for youth-friendly health services, some perinatal adolescents in this study preferred attending antenatal clinics where older women were also service users as they felt they could learn about motherhood from them. Nevertheless, some participants reported their dissatisfaction over attending antenatal clinics where

older women were also service users as they felt judged (Chikalipo et al., 2018).

### Strengths and limitations

To the best of our knowledge, this is the first qualitative study to explore risk and protective factors and care needs for common mental health problems in perinatal adolescents in Malawi. We triangulated evidence from multiple stakeholders thereby providing further depth and credibility to the findings. In addition, the inclusion of participants from both urban and rural settings enabled us to obtain views from perinatal adolescents with diverse cultures and backgrounds, representative of most of the perinatal adolescent population.

Nevertheless, the study has some limitations. Participants were not recruited based on whether they were experiencing mental health problems or had a history of mental health problems. Therefore, they might have been limited in their experiences of mental health care and their responses were based on what they would hypothetically do if they experienced mental health problems. Another limitation is that the study was conducted in government health centres hence the sample might not be representative of other perinatal women attending private hospitals. Due to limited time and resources, we did not return the transcripts to the participants for comments or corrections. In addition, participants did not provide feedback on the findings of the study to ensure the validity of the results.

### Conclusion and practice implications

The findings from this study have some key implications in regard to interventions for common mental health problems (particularly depression and anxiety) among perinatal adolescents. First, interventions need to go beyond targeting symptoms of depression and anxiety to address wider contextual risk factors while enhancing protective factors at different socioecological levels. Also, considering the scarcity of mental health specialists, current healthcare systems need to be strengthened by training non-specialist health workers who provide maternal care in the detection and management of perinatal mental health problems. Considering that perinatal adolescents are at high risk of common mental health problems, such as depression and anxiety, integrating screening of common perinatal mental health problems into routine antenatal clinics can help improve detection. Although it may be argued that doing this may increase the workload of the healthcare workers, a recent study (Chorwe-Sungani et al., 2022a) found that it is feasible and acceptable to implement a Screening Protocol for Antenatal Depression (SPADe), delivered by midwives, in routine antenatal clinics in Malawi, especially if the screening tools are brief. In addition, providing information about mental health during routine antenatal clinics may destigmatise perinatal common mental health problems, thereby encouraging help-seeking. A subsequent participatory co-design study will focus on exploring stakeholders' priorities and preferences for a scalable intervention. The results of this study will lead to the development of an intervention "blueprint".

**Open peer review.** To view the open peer review materials for this article, please visit http://doi.org/10.1017/gmh.2023.64.

**Data availability statement.** Due to conditions of participant consent, the data from this study are not publicly available.

**Acknowledgements.** We wish to express our gratitude to all the participating perinatal adolescents, adolescents' family members, and healthcare workers at Naisi and Matawale primary health centres for their time and contributions.

**Author contribution.** W.M., D.M. and D.G. contributed to the conception and design of the study, data analysis and interpretation. W.M. collected data for the study and led the drafting of the manuscript. D.M. and D.G. critically revised the manuscript. All authors read and approved the final manuscript.

**Financial support.** This work was funded by the Commonwealth Scholarship Commission and awarded to WM, a PhD candidate at the University of Sussex, UK. The funder had no role in the study design, data collection, data analysis, data interpretation, and writing of the manuscript.

**Competing interest.** The authors declare no competing interests exist.

**Ethics standard.** Ethical approvals were obtained from the Sciences & Technology Cross-Schools Research Ethics Committee at the University of Sussex (ER/WM90/1) and the University of Malawi Research Ethics Committee (Protocol No. P.11/21/103).

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
