## [Reviewer Report]

Ms. Wezi Mhango

School of Psychology

University of Sussex

Falmer, Brighton, BN1 9QF

Wm90@sussex.ac.uk

Professor Judith Bass and Dr. Dixon Chibanda 

Co-Editors-in-Chief, Cambridge Prisms: Global Mental Health

22 March 2023

Dear Prof. Bass and Dr. Chibanda 

RE: “I felt I needed help, but I did not get any”: A multiple stakeholder qualitative study of risk and protective factors, and barriers to addressing nkhawa (“worry”) among perinatal adolescents in Malawi

I am pleased to submit the above-named manuscript for consideration as an original research paper in Cambridge Prisms: Global Mental Health. I can confirm the manuscript is not under review with any other journal and we have no conflicts of interest.

This qualitative study is the first phase of a larger programme of study aimed to develop a contextually relevant psychosocial intervention for perinatal adolescents in Malawi. It investigated the risk and protective factors for perinatal depression and anxiety among adolescents in Malawi, and the barriers to accessing care through current service provision. The findings, drawing on qualitative interviews with 26 participants from three stakeholder groups (perinatal adolescents, their family members, and healthcare workers), demonstrate a variety of contextual challenges faced by adolescents in the perinatal period in Malawi at individual, community, and organisational (healthcare systems) levels. The need to strengthen health systems to effectively integrate mental health into the usual maternal health care has also been emphasised. The study underscores the importance of developing a transdiagnostic psychosocial intervention that goes beyond targeting symptoms of depression and anxiety to addressing the wider contextual risk factors while enhancing protective factors at different socioecological levels. 

Please do not hesitate to get in contact for any further information.

Sincerely,

Wezi Mhango (on behalf of all co-authors)

PhD Candidate, School of Psychology, University of Sussex

Corresponding author

---

## [Reviewer Report]

Review 28 May 2023

Thank you for inviting me to review this paper. The research addresses a crucial and under-examined problem in low-income settings – that of distress in adolescents in the perinatal period. There is a paucity of data from Malawi on this problem and a pressing need to generate and disseminate this information given the extraordinarily high levels of adolescent pregnancy in that country.

The researchers have conducted a thoughtful study and I recommend publication with minor changes.

My main question relates to the purpose or added value of conducting mental health screening for the perinatal adolescent study participants. There are three reasons I feel this was unnecessary.

1. the authors oriented the participants to symptoms of depression and anxiety through the use of vignettes

2. the screening tools used are not necessarily valid (see below)

3. there did not seem to be any analysis or discussion of the screening findings in relation to the findings in the framework analysis.

Furthermore, with regard to ethical procedures, the authors should have declared how they supported participants with varying levels of distress as indicated by the screening scores.

With respect to COREQ reporting, the paper lacks mention of 1) Reflexivity of the research team and 2) aside from two coders agreeing on the same analytic framework from coding the same three transcripts – there is no mention of other measures taken to enhance rigor or ensure reliability and validity.

Impact statement:

• For several reasons, I feel that the term ‘unborn child’ is not appropriate for an academic journal. I suggest the term ‘fetus’ be used instead.

• The final sentence refers to integrated mental health screening into routine antenatal care ‘can help improve detection’. However, this study did not set out to establish this, nor did the findings necessarily support this statement.

Background:

• Line 100 – suggest you include ‘any published evidence of’ after ‘find’ as there may well be examples of these interventions that were not picked up in the review cited. Perhaps indicate that the review was systematic.

Methods:

• Eligible perinatal adolescent participants are stated here as those 19 years old and younger. The abstract states the eligible age range was 16-19 years. I suggest you clarify. There are ethical considerations in sampling children under 16 years that would need to be fully elucidated.

• The use of screening tools elucidates symptoms of depression and symptoms of anxiety and does not provide a diagnosis of these conditions. Thus, the authors should be cautious about their use of language in this regard. Moreover, even though those tools (EPDS, GAD and SRQ), originating from the global north, have been translated into Chichewa, there is no published psychometric data to ascertain validity. Additionally, the tools do not reflect local idioms of distress which further limits their validity. This would need to be reflected in the limitations section.

• The use of case vignettes to orientate participants to the problems of depression and anxiety is commendable. It would be relevant to state whether or not these vignettes reflected characters that were perinatal adolescents. If not, this may be construed as a limitation given that perinatal adolescents may present with different clinical features.

Results:

• I suggest replacing the term ‘delivery’ with ‘birth’ which recognises and the woman’s active role in the process of labour and delivery.

• Lines 306-309 is a long run-on sentence which is slightly confusing.

• Under the sub-theme ‘barriers to maternal health care’, the example of lack of access to medication is perhaps of less relevance than the examples of myths and personal attitudes – given the focus of this work on mental health.

• Lines 371-373 – suggest rewording to avoid repetition. It would be useful to note more specifically which staff cadres the ‘clinicians’ comprise of, given the implications for potential recommendations for service design. The term ‘clinical officer’ may need to be explained for some readers.

• Is the term ‘witchdoctor’ commonly used in English in Malawi or would a local term be more appropriate to connote traditional health practitioners or herbalists?

• Line 393 – plural for ‘adolescents’

• Suggest avoiding informal language such as ‘kids’ and ‘one-off’.

Discussion:

• Line 440 - the statement that low self-esteem usually results from body changes of pregnancy is overly reductionist and does not acknowledge the wide array of factors that may contribute to low self-esteem in perinatal adolescents.

• Line 448 – suggest changing the term ‘the man responsible for the pregnancy’

• It should be made explicit what the term ‘most adolescents’ refers to. Is this most adolescents in the participant group or is this most adolescents as reported by all participant groups. During the discussion, this is sometimes clear, but on occasion, it is not.

• Line 475 – what is an HSA? This acronym has not previously been used in the paper (other than the term in full being used in the table).

---

## [Reviewer Report]

Title

1. There is a need to distinguish between depression and anxiety which have been referred to as “nkhawa” in this study.

i. Depression as a symptom can be reffered to as “kukhumudwa” while depression as a disorder is referred to as “Matenda okhumudwa”

ii. Similarly, anxiety as a symptom can be referred to as “Nkhawa” while anciety as a disorder is referred to as “Matenda a nkhawa”

iii. These two concepts have not been properly contextualized in this study and may mislead the reader and other researchers. Revise

Introduction

1. Include the prevalence of perinatal depression in Malawi in paragraph 1 line 81. Read “Chorwe-Sungani, G., Wella, K., Mapulanga, P., Nyirongo, D., & Pindani, M. (2022). Systematic review on the prevalence of perinatal depression in Malawi. South African Journal of Psychiatry, 28(1), 1-7.”

Methods

2. Justify why cross-sectional qualitative study design was suitable for this study

3. Justify why the setting was suitable for this study in relation to perinatal depression and anxiety

i. Provide adequate details for sample size as to why a sample size of 26 was considered adequate for this study.

ii. Clarify why consecutive sampling was suitable for this study?

iii. Specify a sampling frame that was used in this study.

4. Provide details about validity of EPDS, GAD 7 and SRQ 20 in the local context

Results and Discussion

5. Line 434, “…..lack of social support….” This was also found as a risk factor in Malawi and you need to also cite Chorwe-Sungani, G., & Chipps, J. (2018). A cross-sectional study of depression among women attending antenatal clinics in Blantyre district, Malawi. South African Journal of Psychiatry, 24.

---

## [Reviewer Report]

14th August, 2023

Dear Dr. Chibanda,

Re: Revised manuscript, “I felt I needed help, but I did not get any”: A multiple stakeholder qualitative study of risk and protective factors, and barriers to addressing common mental health problems among perinatal adolescents in Malawi

We are grateful for the opportunity to revise and resubmit the above-named manuscript. We are appreciative of the positive feedback offered by the peer reviewers, as well as their thoughtful suggestions on potential improvements. 

We have responded to each of the reviewers’ specific comments in turn. These have been addressed in the same order that they appeared in the original reviews. Numbers have been added against specific comments for ease of reference. Changes have additionally been highlighted in grey in the resubmitted manuscript. Due to the nature of the revisions and clarifications suggested, the word count has increased from 4998 to 5908.

We trust that these revisions prove satisfactory and look forward to your response in due course.

Yours sincerely,

Wezi Mhango

---

## [Reviewer Report]

The study presented in this academic paper explores the critical issue of perinatal depression and anxiety among adolescents, exploring contributing and protective factors as well as service-related factors. The paper provides novel information on a topically relevant problem in Malawi and in many other resource constrained settings. It is important formative work towards the development of a culturally appropriate psychosocial intervention. The research offers actionable insights with important implications for practice and policy in Malawi and similar settings.

I have mostly minor suggestions or comments for consideration for improving the paper.

Introduction

• Line 97 – the lack of implementation of services is likely to be as much about factors such as: lack of resources, lack of knowledge or will of health planners/managers/ministry, competing health priorities, stigma etc. – as it is about lack of evidence.

• Lines 109-110 – as health or social service providers are typically involved in the management of mental health conditions, the phrase on using local idioms of distress surely pertains equally to these groups as much as to researchers (although the same person can be both). I would argue that use of local idioms of distress may particularly be relevant for detection or screening activities.

Setting

• From the description of the participants, it seems that psychosocial services are provided at the research sites (line 155). Given the focus of the investigation, it would be important briefly to describe these services for context as well as some detail of the other services provided at these sites and whether the services are typical for Zomba or Malawi, more broadly.

Participants

• Was there a lower age limit for eligibility of the perinatal adolescents? This would be associated with ethical considerations. I note that the youngest adolescent participant was 16. However, it would be important for the authors to have pre-emptively considered the lower age limit.

Measures

• Information of the local validity of the screening tools should be provided.

Procedures

• What procedures were put in place for participants who endorsed symptoms of depression or anxiety This is ethically important.

Results

• Most of the descriptors for the adolescent participants generating the quotes are numbered, but some are not. I suggest consistently using numbers throughout.

• Lines 263 – grammatical error

• Lines 306-309 – suggest rephrasing for grammar and to avoid a run-on sentence.

• Line 371 – could you describe what cadre of provider the clinicians are? This is relevant to the reader understanding the possibilities of mental health care within the service. Is the clinical officer linked to the quote of lines 374-376 a mental health officer, another type of officer or a generalist? This relates to my earlier comment about orientating the reader to existing services and service provider cadres. Further, it would be useful to know about the location of the different providers with respect to the location of the maternity service.

• Line 384 – is the term ‘witch doctor’ commonly used in Malawi? The phrase has derogatory connotations in many settings, and more specific terminology may be better, e.g. traditional healers or herbalists?

• Line 393 – typo adolescents plural

• Line 396 – the term ‘self-help’ connotes a useful response and some of the examples given are not necessarily adaptive. I suggest using more neutral term to describe these responses, e.g. coping strategies.

• Regarding the subtheme on supply side barriers

o It would be good to know whether the opinion that all maternity care workers should be capacitated to provide mental healthcare was shared by other participants.

o If there were any other data emerging for this theme, it would be good for this to be reported given the formative purpose of this research towards the development of an intervention.

Discussion

• Line 463 – it is not clear from the results section that adolescents ‘recognised that the hospital may be the best place to seek mental health care’

General

• It is stated that data is presented according to COREQ checklist criteria, yet several important items were not included; namely several items in domain 1 “research team and reflexivity”, non-participation, data saturation, transcript checking with participants, use of data software, participant checking.

• Although gendered factors emerged in the findings, matters relating to violence against women are notably absent in this paper. This is problematic given that it is one of the strongest predictors of perinatal mental health conditions and may be particularly prevalent among adolescents. Did the authors explore this with respect to risk factors or with respect to potential barriers to uptake of care? You may be interested in the following papers;

o Chasweka, R., 2018. Isn’t pregnancy supposed to be a joyful time? A cross-sectional study on the types of domestic violence women experience during pregnancy in Malawi. Malawi Medical Journal, 30(3), pp.191-196.

o Brar, S.K., Beattie, T.S., Abas, M., Vansia, D., Phanga, T., Maseko, B., Bekker, L.G., Pettifor, A.E. and Rosenberg, N.E., 2020. The relationship between intimate partner violence and probable depression among adolescent girls and young women in Lilongwe, Malawi. Global Public Health, 15(6), pp.865-876.

o Chilanga, E., Collin-Vezina, D., Khan, M.N. and Riley, L., 2020. Prevalence and determinants of intimate partner violence against mothers of children under-five years in Central Malawi. BMC Public Health, 20, pp.1-14.

• It is a pity that the researchers seemed not to include enquiry about possible models or approaches for mental healthcare provision for their participants to consider. This information would have been particularly useful to inform intervention design. Given the lack of existing mental healthcare offerings in the research setting, and the lack of experience of what mental healthcare may look like for all participant groups, asking about acceptability and feasibility of specific provision options would have yielded useful data. Perhaps this was indeed investigated and forms the content of a separate paper?